# The Neurokinin-1 Receptor Is a Target in Pediatric Rhabdoid Tumors

**Julian Kolorz** [1,†], **Salih Demir** [1,†], **Adrian Gottschlich** [2], **Iris Beirith** [3], **Matthias Ilmer** [3,4], **Daniel Lüthy** [1], **Christoph Walz** [5], **Mario M. Dorostkar** [6], **Thomas Magg** [7], **Fabian Hauck** [7], **Dietrich von Schweinitz** [1], **Sebastian Kobold** [2,4,8], **Roland Kappler** [1] and **Michael Berger** [1,*]

1. Research Laboratories, Department of Pediatric Surgery, Dr. von Hauner Children's Hospital, Ludwig-Maximilians-University Munich, 80337 Munich, Germany; julian.kolorz@med.uni-muenchen.de (J.K.); salih.demir@med.uni-muenchen.de (S.D.); daniel.luethy@med.uni-muenchen.de (D.L.); dietrich.schweinitz@med.uni-muenchen.de (D.v.S.); roland.kappler@med.uni-muenchen.de (R.K.)
2. Center for Integrated Protein Science Munich (CIPSM) and Division of Clinical Pharmacology, Department of Medicine IV, University Hospital, Ludwig-Maximilians-University Munich, 80337 Munich, Germany; adrian.gottschlich@med.uni-muenchen.de (A.G.); sebastian.kobold@med.uni-muenchen.de (S.K.)
3. Department of General, Visceral, and Transplantation Surgery, University Hospital, Ludwig-Maximilians-University Munich, 81377 Munich, Germany; iris.beirith@med.uni-muenchen.de (I.B.); matthias.ilmer@med.uni-muenchen.de (M.I.)
4. German Center for Translational Cancer Research (DKTK), Partner Site Munich, 81377 Munich, Germany
5. Institute of Pathology, Faculty of Medicine, Ludwig Maximilians-University Munich, 80337 Munich, Germany; christoph.walz@med.uni-muenchen.de
6. Center for Neuropathology, Ludwig-Maximilians-University Munich, 81377 Munich, Germany; mario.dorostkar@med.uni-muenchen.de
7. Department of Pediatrics, Dr. von Hauner Children's Hospital, Ludwig-Maximilians-University Munich, 80337 Munich, Germany; thomas.magg@med.uni-muenchen.de (T.M.); fabian.hauck@med.uni-muenchen.de (F.H.)
8. Einheit für Klinische Pharmakologie (EKLiP), Helmholtz Zentrum München, German Research Center for Environmental Health (HMGU), 85764 Neuherberg, Germany
* Correspondence: michael.berger@med.uni-muenchen.de; Tel.: +49-89-4400-57859
† These authors contributed equally to this work.

**Abstract:** Rhabdoid tumors (RT) are among the most aggressive tumors in early childhood. Overall survival remains poor, and treatment only effectively occurs at the cost of high toxicity and late adverse effects. It has been reported that the neurokinin-1 receptor/ substance P complex plays an important role in cancer and proved to be a promising target. However, its role in RT has not yet been described. This study aims to determine whether the neurokinin-1 receptor is expressed in RT and whether neurokinin-1 receptor (NK1R) antagonists can serve as a novel therapeutic approach in treating RTs. By in silico analysis using the cBio Cancer Genomics Portal we found that RTs highly express neurokinin-1 receptor. We confirmed these results by RT-PCR in both tumor cell lines and in human tissue samples of various affected organs. We demonstrated a growth inhibitory and apoptotic effect of aprepitant in viability assays and flow cytometry. Furthermore, this effect proved to remain when used in combination with the cytostatic cisplatin. Western blot analysis showed an upregulation of apoptotic signaling pathways in rhabdoid tumors when treated with aprepitant. Overall, our findings suggest that NK1R may be a promising target for the treatment of RT in combination with other anti-cancer therapies and can be targeted with the NK1R antagonist aprepitant.

**Keywords:** rhabdoid tumor; NK-1 receptor; NK-1 receptor antagonist; substance P; cancer; apoptosis

## 1. Introduction

Rhabdoid tumors (RT) are rare and highly aggressive tumors primarily affecting infants and young children [1–3]. They have been reported to be located in several organ compartments, such as the central nervous system (CNS) (referred to as atypical

teratoid/rhabdoid tumor [AT/RT]), kidneys (RT of the kidney [RTK]), the liver, and soft tissue (extrarenal RT, malignant RT [MRT]) [1,3–7]. One of the genetic hallmarks of RTs is a loss-of-function mutation of the SWI/SNF-related matrix-associated actin-dependent regulator of chromatin subfamily B member 1 (SMARCB1), also named integrase interactor 1 (INI1). INI1 acts as a component of the SWItch/Sucrose Non-Fermentable (SWI/SNF) chromatin-remodeling complex, which functions as a tumor suppressor [1,8]. Prognosis of children with RT has improved, but overall survival remains unsatisfactory with less than 50% for AT/RTs and less than 40% for MRTs [9–12]. The lack of standard effective therapy, considerable concern about the toxicity of the chemotherapeutics, and late adverse effects require an improvement in the treatment of RT [1,2,9,12–14].

The involvement of the neurokinin-1 receptor (NK1R; *TACR1*)/substance P (SP; *TAC1*) complex in cancer has been described previously [15–19]. By binding to NK1R, SP promotes a variety of functions to improve the growth and survival of tumor cells [20]. Thus, the use of NK1R antagonists can serve as a desirable target for cancer treatment.

Two isoforms of NK1R have been reported. Full-length (fl-) NK1R contains 407 amino acids, whereas truncated (tr-) NK1R only consists of 311 amino acids, lacking 96 amino acids at the cytoplasmic C-terminus of the receptor [19,21]. It has been demonstrated that the truncated isoform is expressed higher in the tumor in comparison to the full-length isoform [17]. Furthermore, the full-length form has been associated with slow growth of cells, whereas an upregulation of the truncated version has been related with rapid growth and more aggressive behavior of tumor cells [22,23].

The non-peptide NK1R antagonist aprepitant is approved by the *Food and Drug Administration (FDA)* for the treatment of chemotherapy-induced nausea and vomiting. However, it has been shown to exert antiemetic, antipruritic, antiviral, and a broad variety of antitumor actions as well [20]. Importantly, the side effects of aprepitant are minimal, and even high doses do not seem to influence the proliferative capacity of healthy cells [24–26].

Until now, the role of the NK1R/SP complex in rhabdoid tumors remained unknown. For the first time, we describe the expression of NK1R and its truncated splice variant in rhabdoid tumors, and that it can be targeted with NK1R antagonist aprepitant, serving as a novel target for the treatment of RT.

## 2. Materials and Methods

### 2.1. Cell Culture

Two AT/RT cell lines, BT-12 and CHLA-266, one RTK cell line, G-401, and one hepatoblastoma (HB) cell line, HepG2, were used throughout this study. The two cell lines BT-12 and CHLA-266 were obtained from the Childhood Cancer Repository of the Children's Oncology Group at Texas Tech University Health Sciences Center (Lubbock, TX, USA). Both AT/RT cell lines were grown in Iscove's Modified Dulbecco's Medium (Gibco, Carlsbad, CA, USA) supplemented with 20% heat-inactivated fetal calf serum (FCS) and 100 μg/mL streptomycin 100 U/mL penicillin. G-401 was purchased from the ATCC (Manassas, VA, USA) and was grown in McCoy-5A-Medium (ATCC, Manassas, VA, USA) supplemented with 10% FCS and 100 μg/mL streptomycin 100 U/mL penicillin. HepG2 was grown in RPMI (Gibco, Carlsbad, CA, USA) supplemented with 10% FCS and 100 μg/mL streptomycin 100 U/mL penicillin as previously described [19,27]. The two primary dermal fibroblasts PCS-201-012 (Adult) and PCS-201-010 (Neonatal) were both obtained from the ATCC (Manassas, VA, USA) and grown in DMEM (Gibco, Carlsbad, CA, USA) with 10% FCS and 100 μg/mL streptomycin 100 U/mL penicillin. All cells were grown at 37 °C in a humidified incubator with 5% $CO_2$. Mycoplasma contamination was regularly excluded using mycoplasma-specific polymerase chain reactions protocols.

### 2.2. Drugs

The NK1R antagonist aprepitant and the cytostatic compound cisplatin were purchased from Selleck Chemicals (Houston, TX, USA) and were dissolved in DMSO. SP

(NK1R agonist) was purchased from Sigma-Aldrich (St. Louis, MO, USA) and dissolved in 0.1 mol/L acetic acid and purified water.

### 2.3. Proliferation Assays (MTT Assay)

MTT (3-(4,5-dimethylthiazol-2-yl)-2,5-diphenyltetrazolium bromide) salt was purchased from Sigma-Aldrich (St. Louis, MO, USA) and dissolved in PBS (MTT solution; 5 mg/mL). $5 \times 10^4$ cells/well were seeded into a 96-well plate and were incubated overnight in culturing media for attachment. The cells were exposed to 10 different increasing concentrations of the corresponding compounds (from 0.02 μM to 100 μM) for 48 h. For combination treatment, the cells were exposed to the corresponding compounds for 48 h. After exposure, culturing media was replaced with the MTT solution for 4 h at 37 °C. Then, MTT solution was replaced with 10% SDS in 0.01 M HCl for overnight incubation in the incubator. 96-well plates were measured using a FLUOstar Omega microplate reader (BMG LABTECH Inc., Cary, NC, USA) at 595 nm wavelength.

### 2.4. In Vitro Analysis of Apoptosis

The determination of apoptotic cell populations was performed by flow cytometry. Annexin V staining was performed according to manufacturer's instructions. $5 \times 10^4$ cells/well were seeded into a 96-well plate. The cells were incubated overnight in the culturing media for attachment and then were exposed to the corresponding compounds for 48h. Cells were stained with 2 μL Pacific Blue Annexin V (BioLegend, San Diego, CA, USA) for 15 min at RT in dark. Annexin V-binding buffer ($10\times$ Binding Buffer: 10 mM HEPES/NaOH, 140 mM NaCl, 2.5 mM CaCl2) was used for respective staining and washing steps. Fixable Viability Dye eFluor$^{TM}$ 780 (FVD, eBioscience, Inc., San Diego, CA, USA) was used to discriminate viable and dead cell populations. Staining was carried out at 4 °C for 30 min in the dark. Percentages of viable, apoptotic and necrotic cells were measured with a BD FACSCanto II (BD Biosciences, Franklin Lanes, NY, USA) and results were analyzed using FlowJo 10.0 Software (Tree Star, Inc., Ashland, OR, USA).

### 2.5. Western Blot Analysis

Protein expression of PARP-1 (rabbit, 1:1000 dilution; Cell Signaling Technologies, Danvers, MA, USA) and α-tubulin (mouse, 1:5000 dilution; Sigma-Aldrich, St. Louis, MO, USA) was analyzed by Western blot analysis. Cells were lysed for 30 min on ice with lysis buffer (tris-HCL 30 mM, NaCl 150 mM, tritonX 1% and glycerol 10%) and then underwent high-speed centrifugation. Protein concentration was assessed by Bradford assay (Bio-Rad Laboratories, Hercules, CA, USA). 20 mg protein per well was separated by Novex WedgeWell 4 to 20%, Tris-Glycine, 1.0 mm, mini protein gels (Invitrogen, Carlsbad, CA, USA) and electroblotted onto 0.2 μm PVDF membranes using Trans-Blot Turbo mini transfer packs (Bio-Rad Laboratories, Hercules, CA, USA). After blocking for 1 h in phosphate-buffered saline supplemented with 5% milk and 0.1% Tween 20 (Sigma-Aldrich, St. Louis, MO, USA), immunodetection was performed using polyclonal goat anti-mouse IgG (P0447; 1:20,000) and polyclonal goat anti-rabbit IgG (P0448; 1:2000) antibodies (both from DakoCytomation Denmark A/S, Glostrup, Denmark). Enhanced chemiluminescence was used for detection (GE Healthcare Amersham™ ECL™ Prime Western blotting detection reagents, Thermo Scientific, Waltham, MA, USA) and imaging was performed at ChemiDoc XRS+ (Bio-Rad Laboratories, Hercules, CA, USA). The size of proteins on Western blots was identified by PageRuler Prestained Protein Ladder (Thermo Scientific, Waltham, MA, USA).

### 2.6. RT-PCR (Reverse Transcription Polymerase Chain Reaction)

RNA extraction, complementary DNA synthesis, and quantitative polymerase chain reaction (qPCR) analysis were performed as previously described [27]. Specific primers were as follows: *TACR1-fl* (NM_001058.3; forward 5′-AACCCCATCATCTACTGCTGC-3′ and reverse 5′-ATTTCCAGCCCCTCATAGTCG-3′), *TACR1-tr* (forward 5′-

CAGGGGCCACAAGACCATCTA-3′ and reverse 5′-ATAAGTTAGCTGCAGTCCCCAC-3′), *TAC1* (forward 5′-AAGCCTCAGCAGTTCTTTGG-3′ and reverse 5′-TCTGGCCATGTCCATAAAGAG-3′) and *TBP* (forward 5′-GCCCGAAACGCCGAATAT-3′ and reverse 5′-CCGTGGTTCGTGGCTCTCT-3′).

*2.7. Patients and Tumor Samples*

A total of 6 tumor specimens were obtained from pediatric patients. Three rhabdoid tumors of the liver and one rhabdoid tumor of the kidney were resected at the Department of Pediatric Surgery, LMU and preserved in liquid nitrogen. Two formalin-fixed and paraffin-embedded AT/RT tumor samples on slides were obtained from the Center for Neuropathology, LMU. RNA from the former samples was extracted using TRIzol reagent (Invitrogen, Karlsruhe, Germany), from the latter using High Pure FFPET RNA Isolation Kit from Roche Diagnostics Deutschland GmbH (Mannheim, Germany) according to manufacturer's instructions. The study protocol was approved by the Committee of Ethics of the LMU (Munich) and written informed consent was obtained from each patient in accordance with the Declaration of Helsinki.

*2.8. Data Base Analysis*

Data sets were derived from the cBio Cancer Genomics Portal (http://cbioportal.org; accessed on 2 December 2021) [28–30]. Five pediatric cancer studies on acute myeloid leukemia (AML; phs000465), acute lymphoblastic leukemia (ALL; phs000464), neuroblastoma (NB; phs000467), rhabdoid tumors (RT; phs0004709), and Wilms tumor (WT; phs000471) were selected from the Therapeutically Applicable Research to Generate Effective Treatments (https://ocg.cancer.gov/programs/target; accessed on 2 December 2021) initiative and the mRNA expression of 2 genes was analyzed (*TACR1*, *TAC1*).

*2.9. Statistical Analysis*

Results are expressed as the mean ± standard error of the mean (SEM). All statistical comparisons were made with an unpaired parametric t-test comparing two groups, an ordinary one-way ANOVA or a Tukey´s multiple comparison test, with a single pooled variance using GraphPad Prism (San Diego, CA, USA). The significance was considered as: $p < 0.05$ (*), $p < 0.01$ (**), $p < 0.001$ (***) and $p < 0.0001$ (****) for all comparisons.

## 3. Results

### 3.1. Expression of TACR1 and TAC1 in Pediatric Cancer

It has been previously demonstrated that the NK1R/SP complex serves as a target in a large variety of cancers, including pediatric malignancies such as neuroblastoma and hepatoblastoma [19,20,31–33]. Nothing was known of the expression of this potent target in rhabdoid tumors. Therefore, as a first step we assessed expression levels for the NK1R/SP complex (TACR1, TAC1) of the most common childhood malignancies from the cBio Cancer Genomics Portal, focusing on acute myeloid leukemia, acute lymphoblastic leukemia, neuroblastoma, and Wilms' tumor, and compared them to the expression levels in rhabdoid tumors [28]. Database analysis showed that the TACR1 gene is expressed significantly higher in rhabdoid tumors in comparison to the aforementioned tumors (Figure 1a). TAC1 mRNA (which codes for SP and is the natural ligand of NK1R) expression levels were high in rhabdoid tumors and neuroblastoma, but not significantly different to the other tumors (Figure 1b). As this in silico analysis did not allow for a sub-analysis between the expression levels of the full-length and truncated splice variant of NK1R, we next determined their gene expression levels in RT cell lines by RT-PCR (Figure 1c–f). The RTK cell line G-401 and the two AT/RT cell lines BT-12 and CHLA-266 are known to carry loss-of-function mutation of INI1 [34,35] and their characteristics are summarized in Table 1. Interestingly, expression levels of TACR1-fl and TACR1-tr genes in G-401, BT-12, and CHLA-266 were similar to HepG2, a human hepatoblastoma (HB) cell line with a known high expression of TACR1-tr [19] used as a positive control. The adult and neonatal

primary dermal fibroblasts expressed significantly less TACR1-tr than G-401 and were used as a negative control (Figure 1e). TACR1-tr was significantly overexpressed in every cell line compared to TACR1-fl (Figure 1c). Then, expression levels of TAC1 were tested in the different cell lines. The expression levels of TAC1 were detected to be very low in all cell lines analyzed except for the primal dermal fibroblasts (Figure 1d). Of the tumor cell lines, G-401 showed the highest expression of TAC1, while HepG2 cells have no detectable expression of TAC1 in mRNA levels.

**Table 1.** Summary of characteristics of cell lines.

| Cell Line | Disease | Origin | INI1-Mutation |
|-----------|---------|--------|---------------|
| BT-12 | AT/RT | Female, 2 months, Caucasian | Loss-of-function |
| CHLA-266 | AT/RT | Female, 30 months, Caucasian | Loss-of-function |
| G-401 | RTK | Male, 3 months, Caucasian | Loss-of-function |
| HepG2 | HB | Male, 15 years, Caucasian | – |

AT/RT—atypical teratoid/rhabdoid tumor; RTK—rhabdoid tumor of the kidney; HB—hepatoblastoma; INI1—integrase interactor 1.

Next, we analyzed the expression levels of TACR1 in six rhabdoid tumor tissue samples by RT-PCR. As rhabdoid tumors can arise at different sites throughout the body, we analyzed tissue samples of three MRTs of the liver, one RT of the kidney, and two AT/RTs. The patient characteristics are summarized in Table 2. The age at the time of diagnosis ranged from 0–127 months (median, 18 months). INI1 alterations were detected in all tested samples. Five out of six tumor samples showed a loss-of-function mutation of INI. Interestingly, one patient (T190) was found to carry an INI1 germ line mutation (Table 2). The respective patient was first diagnosed with an AT/RT, and after initial successful treatment, subsequently developed a RTK. In line with the expression pattern observed in AT/RT and RTK cell lines, splice variant TACR1-tr was expressed higher than TACR1-fl. Expression levels of TACR1-tr were found the highest in an RT of the liver tumor tissue sample (T125II), and the lowest in AT/RT tumor tissue samples (T16, T12IC). TACR1-fl expression was found to be very low in all tumor samples analyzed. Surprisingly, the analysis of TAC1 expression revealed that the RT of the kidney (T190) had the highest expression, whereas the other tumor tissue samples showed almost no expression (Figure 1g,h).

In conclusion, we were able to demonstrate the expression of TACR1 in both RT cell lines and primary patient material, with TACR1-tr being significantly higher expressed than TACR1-fl in all tested samples. Importantly, both cell lines and patient tissue samples originated from different compartments of the body, modelling the heterogenicity of rhabdoid tumors.

### 3.2. Clinical Outcome and Biological Characteristics of Patients with Rhabdoid Tumors

Next, we made use of the pediatric rhabdoid tumor study from The cBio Cancer Genomics Portal to understand in detail whether TACR1 or TAC1 expression in rhabdoid tumors correlates with biological or clinical characteristics [28]. The gene expression patterns of 42 patients were analyzed and correlated to tumor stage, gender, age of diagnosis and overall survival (Table 3, Figure 1i–l). Tumor stages were classified by the American Joint Committee on Cancer [36]. mRNA expression for TACR1 showed similar expression in every stage, whereas mRNA expression for TAC1 was slightly elevated in stages I–II. Gender was distributed similarly for TACR1 and TAC1 (45% male vs. 55% female) and no

significant differences in the mRNA expression levels of the genes were observed. Age of diagnosis was prominently represented within the first year of life for both TACR1 and TAC1 (<6 months, 22.5%; 6–12 months, 37.5%). Overall survival of rhabdoid tumor patients with low expression (*n* = 20) of TACR1 or TAC1 was compared to the patients with high expression (*n* = 19) based on the median of TAC1 or TACR1 expression. For three patients, overall survival data was not available. We could not observe any significant differences between TACR1 or TAC1 low-expressing and high-expressing groups.

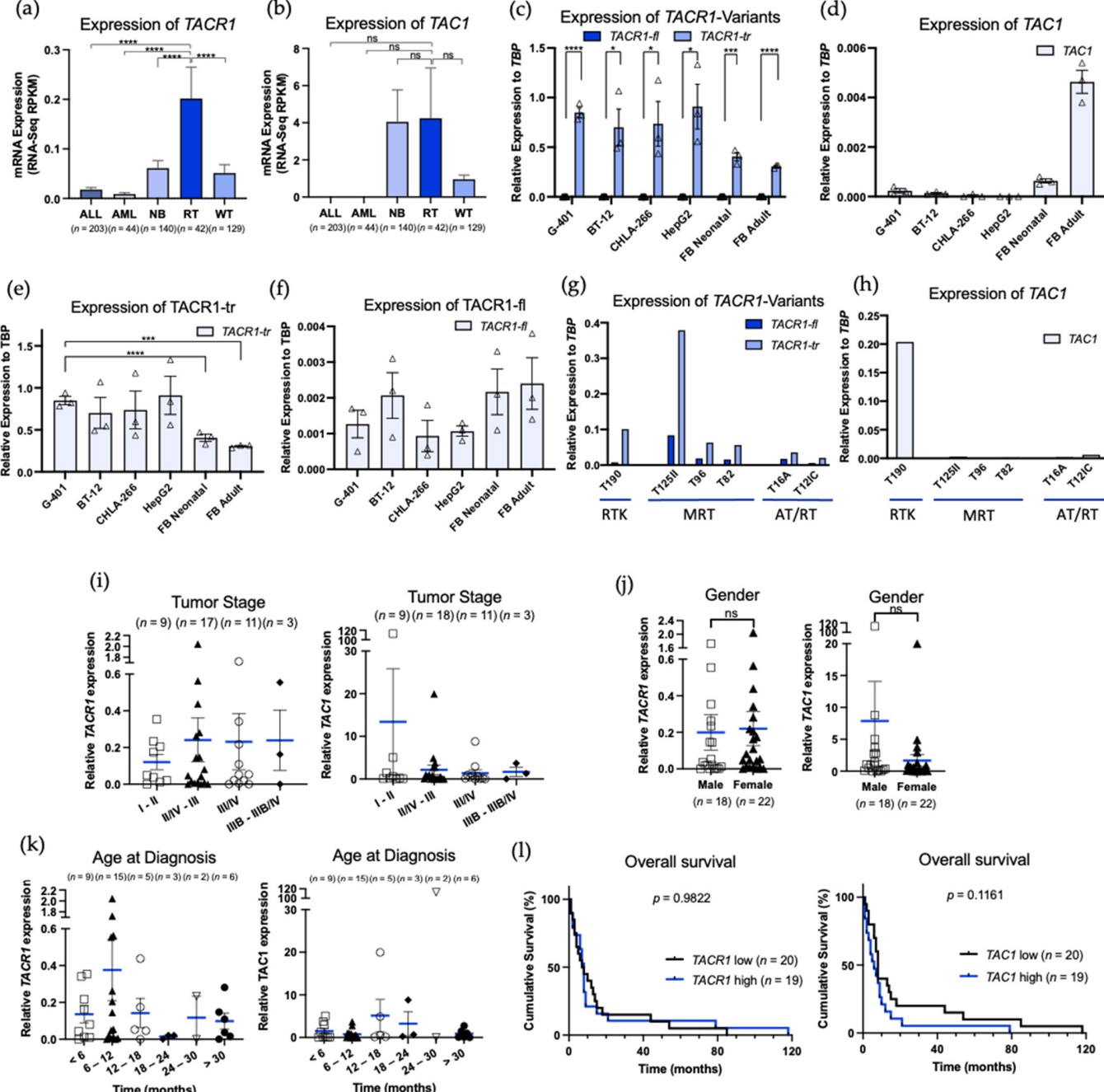

**Figure 1.** Expression of *TACR1* and *TAC1* and biological and clinical parameters of rhabdoid tumors. (**a**,**b**) mRNA expression of *TACR1* and *TAC1* in ALL (*n* = 203), AML (*n* = 44), NB (*n* = 140), RT (*n* = 42), and WT (*n* = 129). Relative expression was correlated to the (**i**) tumor stage by the Neoplasm American Joint Committee on Cancer (I-II *n* = 9; II/IV-III *n* = 17 for TACR1, *n* = 18 for TAC1; III/IV

*n* = 11; IIIB-IIIB/IV *n* = 3), (**j**) gender (Male *n* = 18; Female *n* = 22), (**k**) age at diagnosis (months) (<6 months *n* = 9; 6–12 months *n* = 15; 12–18 months *n* = 5; 18–24 months *n* = 3; 24–40 months *n* = 2; >30 months *n* = 6), (**l**) overall survival (months) (TACR1 low *n* = 20; TACR1 high *n* = 19; TAC1 low (*n* = 20), TAC1 high *n* = 19), and (**c–f**) mRNA expression of *TACR1-fl*, *TACR1-tr,* and *TAC1* in G-401, BT-12, CHLA-266, HepG2, and in FB Neonatal (=fibroblasts neonatal, PCS-201-120) and in FB Adult (=fibroblasts adult, PCS-201-012) normalized to the housekeeping gene *TBP*. (**g,h**) mRNA expression of *TACR1-fl*, *TACR1-tr* and *TAC1* in tumor samples normalized to *TBP*. Results are expressed as the mean ± standard error of the mean (SEM). All statistical comparisons were made with an unpaired parametric t-test comparing two groups, an ordinary one-way ANOVA or a Tukey's multiple comparison test, with a single pooled variance. ns = not significant. $p < 0.05$ (*), $p < 0.001$ (***) and $p < 0.0001$ (****) for all comparisons.

**Table 2.** Summary of the characteristics of tumor samples.

| Case | Name | Subtype of RT (Organ Compartment) | Gender (M/F) | Age at Diagnosis (Months) | INI1-Mutation | Treatment | Relapse |
|---|---|---|---|---|---|---|---|
| 1 | T190 | AT/RT → RTK * | M | <23 * | Germline mutation | Chemotherapy, Resection, Stem-cell therapy | Yes |
| 2 | T125II | MRT (liver) | M | 10 | Loss-of-function | Chemotherapy, Resection | Yes |
| 3 | T96 | MRT (liver) | F | 13 | Loss-of-function | Chemotherapy, Resection, Stem-cell therapy | No |
| 4 | T82 | MRT (liver) | M | 33 | Loss-of-function | Chemotherapy, Resection | No |
| 5 | T16A | AT/RT | M | 5 | Loss-of-function | - | - |
| 6 | T12IC | AT/RT | M | 127 | Loss-of-function | Chemotherapy, Radiotherapy | Yes |

M—male; F—female; RTK—rhabdoid tumor of the kidney; AT/RT—atypical teratoid/rhabdoid tumor; MRT—malignant rhabdoid tumor; INI1—integrase interactor 1; * Relapse.

**Table 3.** Summary of clinical outcome and biological characteristics of the patients from database analysis.

| Characteristics | TACR1 * | TAC1 * |
|---|---|---|
| Tumor Stage | | |
| I–II | 9 (22.5) | 9 (22.0) |
| II/IV–III | 17 (42.5) | 18 (43.9) |
| III/IV | 11 (27.5) | 11 (26.8) |
| IIIB–IIIB/IV | 3 (7.5) | 3 (7.3) |
| Gender | | |
| Male | 18 (45.0) | 18 (45.0) |
| Female | 22 (55.0) | 22 (55.0) |
| Age at Diagnosis (months) | | |
| <6 | 9 (22.5) | 9 (22.5) |
| 6–12 | 15 (37.5) | 15 (37.5) |
| 12–18 | 5 (12.5) | 5 (12.5) |
| 18–24 | 3 (7.5) | 3 (7.5) |
| 24–30 | 2 (5.0) | 2 (5.0) |
| >30 | 6 (15.0) | 6 (15.0) |

* Patients, *n* (%); data sets were derived from the cBio Cancer Genomics Portal.

Overall, we observed that TACR1 and TAC1 were expressed across different tumor stages and independent of gender or age of diagnosis. This conserved expression highlights the functionality of the NK1R/SP axis in rhabdoid tumors independent of clinical or

biological characteristics of the patient. Thus, NK1R redirected targeted therapies would not be restricted to a certain subset of patient suffering from rhabdoid tumors.

### 3.3. Aprepitant Inhibits Tumor Growth in Rhabdoid Tumor Cell Lines and Shows Increased Activity with Cisplatin

To probe the role of NK1R-SP-targeted therapies in rhabdoid tumors, we next investigated the effect of aprepitant on the RT tumor cell lines G-401, BT-12, and CHLA-266 with the human HB cell line HepG2 as a positive control and the primary dermal fibroblasts as a negative control for aprepitant response [19]. Tumor cells were exposed to nine different increasing concentrations of aprepitant (ranging from 10 μM to 100 μM) for 48 h and viability was determined using MTT cell proliferation assay. All RT cells showed a dose-dependent decrease in cell viability after aprepitant treatment in comparison to solvent-treated controls. The neonatal and adult primary dermal fibroblasts showed no strong dose-dependent decrease and were used as a negative control. G-401, BT-12, CHLA-266 and HepG2 cells exhibited similar response towards aprepitant (G-401 and BT-12, 50 μM; CHLA-266, 40 μM; HepG2 15 μM) (Figure 2a).

In order to investigate possible synergism of conventional treatment regimens and NK1R-targeted therapies, we combined aprepitant with cisplatin, a commonly used cytostatic chemotherapeutic agent for the treatment of rhabdoid tumors [1,37]. First, we investigated the viability of the tumor cells upon cisplatin exposure, using MTT assay as described (10 different increasing concentrations, ranging from 0.02 μM to 100 μM). We observed dose-dependent response towards cisplatin in all tumor cells with the exception of the G-401 cells (Figure 2b).

Next, G-401, BT-12, CHLA-266, and HepG2 cells were exposed to aprepitant, cisplatin, and a combination of both (Figure 2c). Lower cisplatin concentrations were used to evaluate possible synergism between the treatments. We observed significant additive effects in HepG2 and the AT/RT cell line CHLA-266. Similar trends were observed for BT-12 and G-401, but the effect was not statistically significant.

In summary, we were able to demonstrate a dose-dependent reduction in the viability of the rhabdoid tumor cell lines G-401, BT-12, and CHLA-266 after treatment with aprepitant and highlight possible additive effects of combinatory treatment approaches combining aprepitant with a conventional cytostatic drug.

### 3.4. Substance P Reverses Anti-Proliferative Effect of Aprepitant

In order to investigate the specificity of aprepitant, we tested whether treatment with the NK1R-ligand substance P could abrogate the observed treatment effects. Thus, G-401, BT-12, and CHLA-266 cells were exposed to a saturating concentration of substance P (200 nM, 48 h). As aprepitant acts as a competitive inhibitor of NK1R [38], high dose treatment with SP should prevent the binding of aprepitant to NK1R, diminishing the treatment effect as previously described for other tumors, including hepatoblastoma [19]. Our experiments revealed that substance P does not cause any toxicity for the cells, as no changes in cell viability were observed (Figure 2d). Importantly, aprepitant and substance P co-exposure significantly inhibited the anti-proliferative effect of aprepitant in BT-12 (Figure 2d). A similar trend was observed for G-401 and CHLA-266, however the effect was not statistically significant.

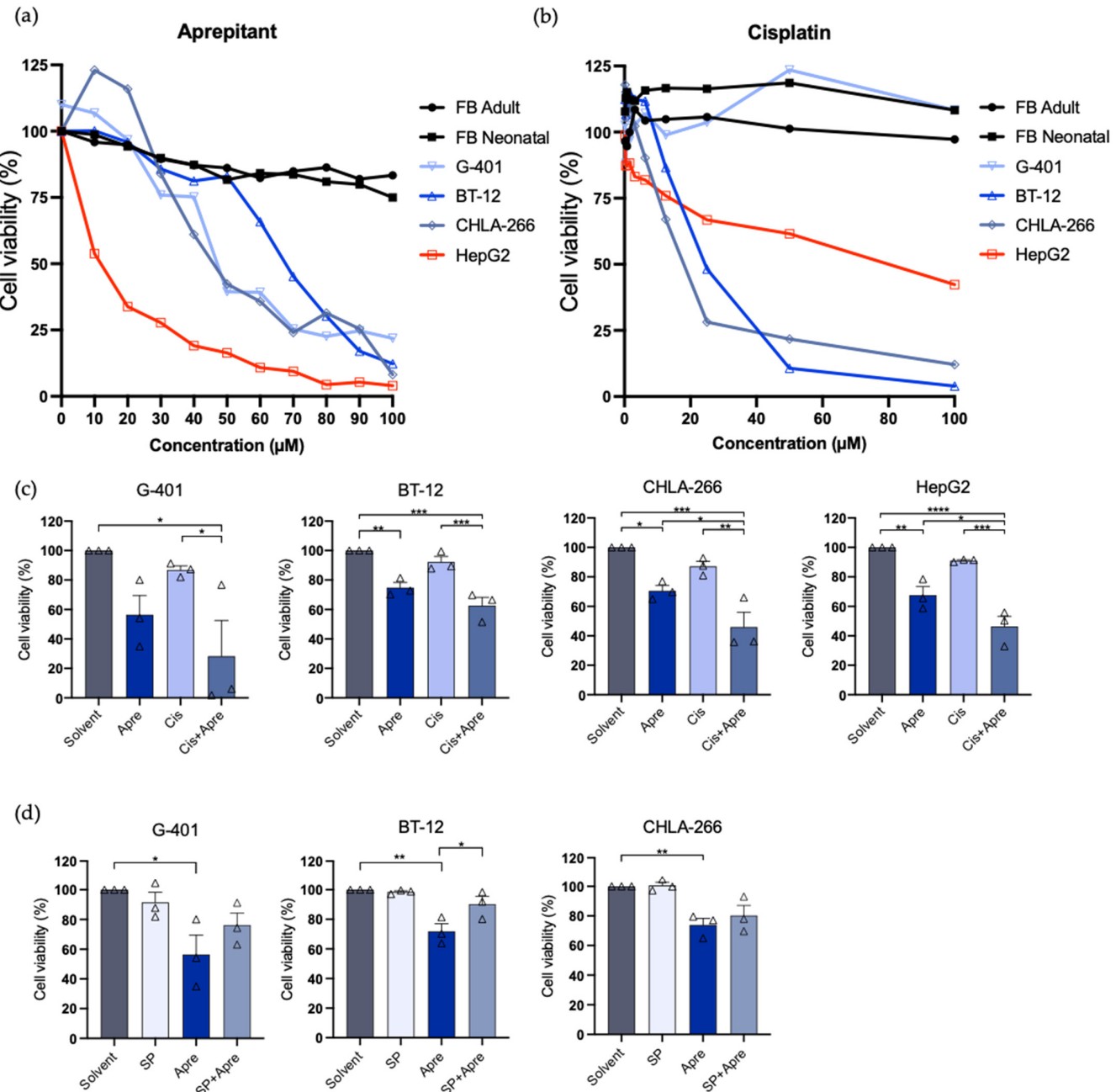

**Figure 2.** Cell proliferation analysis of G-401, BT-12, CHLA-266, HepG2, FB Neonatal, and FB Adult upon treatment with different compounds. (**a**) MTT assay measuring cell viability upon treatment with aprepitant (10–100 μM) for 48 h. (**b**) MTT assay measuring cell viability upon treatment with cisplatin (0.02–100 μM) for 48 h. (**c**) Combination MTT assay measuring cell viability upon treatment of aprepitant (aprepitant concentrations for G-401 and BT-12, 50 μM; CHLA-266, 40 μM; HepG2 15 μM), cisplatin (Cis, 20 μM) and aprepitant + cisplatin (Combi) for 48 h. (**d**) MTT assay measuring cell viability upon treatment of aprepitant (G-401 and BT-12, 50 μM; CHLA-266, 40 μM; HepG2 15 μM) and stimulation with Substance P (SP, 200 nM) and combination of Apre and SP. Pooled data of three independent experiments. Results are expressed as the mean $\pm$ standard error of the mean (SEM). All statistical comparisons were made with an ordinary one-way ANOVA or a Tukey's multiple comparison test, with a single pooled variance. $p < 0.05$ (*), $p < 0.01$ (**), $p < 0.001$ (***) and $p < 0.0001$ (****) for all comparisons.



In conclusion, treatment with a supramaximal dosage of substance P reversed the observed therapeutic effect of aprepitant, highlighting the specificity of NK1R-targeted therapies and emphasizing the role of the NK1R/SP complex as a pro-tumorigenic signaling pathway in rhabdoid malignancies.

### 3.5. Aprepitant Triggers Apoptosis Signaling in Rhabdoid Tumor Cell Lines

To investigate the mechanism of cell death induced by aprepitant in tumor cells, we focused on important effector cascades that are known to take part in cell death induction.

First, we carried out Annexin V staining by using flow cytometry. We observed reduction of viable cell populations upon exposure to aprepitant, comparable to previous experiments. Moreover, we observed increasing fractions of early and late apoptotic cells in rhabdoid tumor cells after 48 h exposure to aprepitant, suggesting apoptosis as the mechanism of aprepitant-induced-cell death. (Figures 3a,b, S1 and S2). Treatment with cisplatin was used as a positive control. Furthermore, a combination of both compounds showed similar effect of increasing apoptotic cell populations.

To confirm these results, we used immunoblotting to analyze the expression of the key pro-apoptotic effector protein poly (ADP-ribose) polymerase 1 (PARP-1). Whole cell lysates from rhabdoid tumor cells were isolated after treatment with aprepitant, cisplatin, or the combination of both. Interestingly, we observed strong expression of the cleaved products of PARP-1 in rhabdoid cells upon aprepitant exposure, in line with the results obtained from flow cytometric Annexin V staining (Figures 4a,b, S3 and S4).

Altogether, these findings show an upregulation of apoptotic signaling pathways in rhabdoid cells when treated with aprepitant, emphasizing the potential efficacy of an aprepitant treatment in RTs.

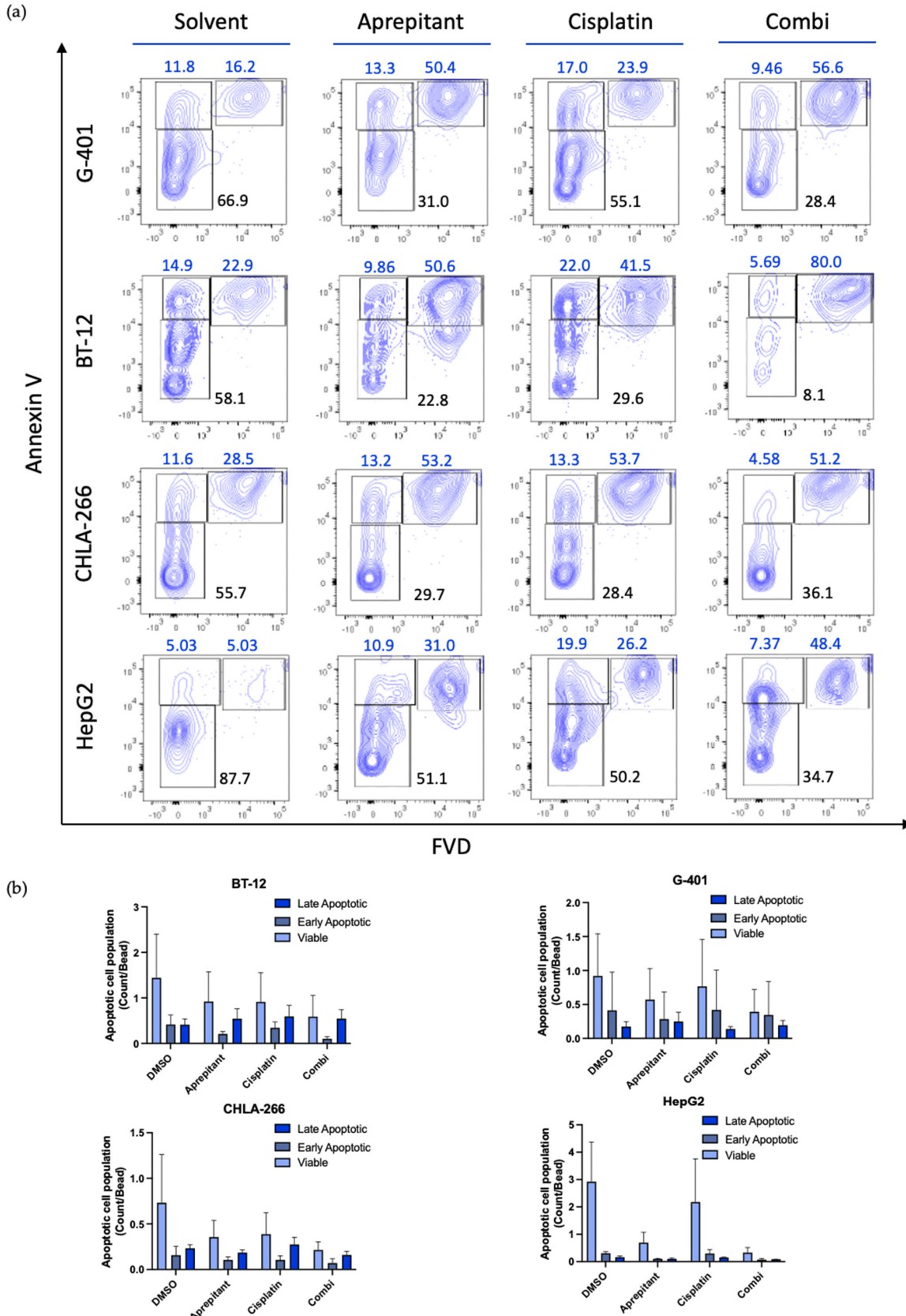

**Figure 3.** (**a**) Determination of apoptotic cell populations of G-401, BT-12, CHLA-266, and HepG2 was performed by staining with annexin V and fixable viability dye (FVD) and assessed through

fluorescence-activated cell sorting. Cells were treated with aprepitant (aprepitant concentrations for G-401 and BT-12, 50 μM; CHLA-266, 40 μM; HepG2, 15 μM), cisplatin (Cis, 20 μM) and aprepitant + cisplatin (Combi) for 48 h. DMSO was used as treatment control. Shown is one representative of three experiments. Numbers represent the percentages of the cell populations; black: viable; blue: apoptotic. (**b**) Quantification of viable and apoptotic cell populations. Shown are results from three independent experiments (*n* = 3).

(a)

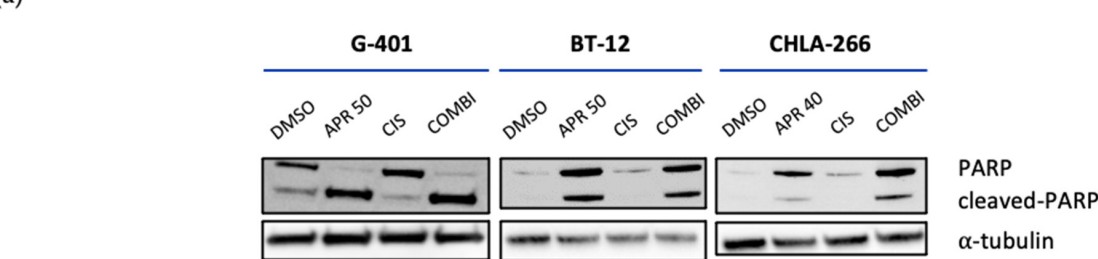

(b)

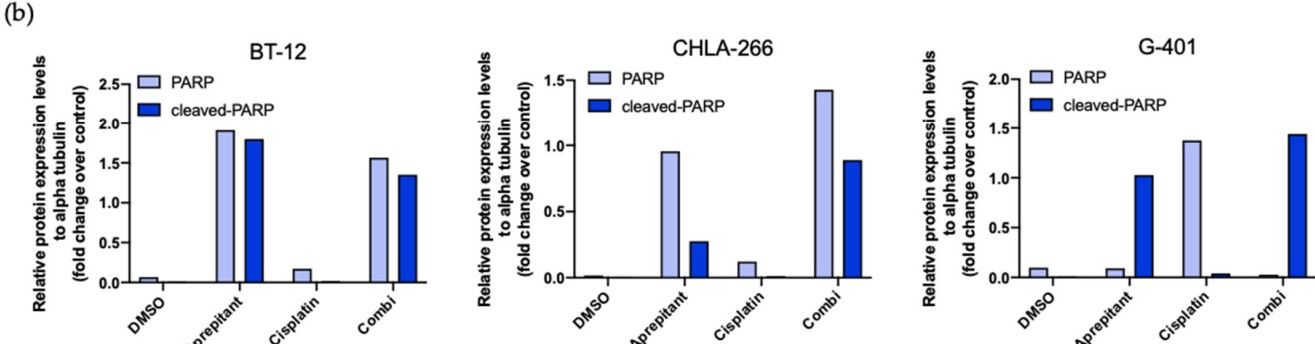

**Figure 4.** (**a**) Western blot analysis of G-401, BT-12 and CHLA-266 upon treatment with aprepitant (G-401 and BT-12, 50 μM; CHLA-266, 40 μM), cisplatin (CIS 20 μM) and aprepitant + cisplatin (COMBI) (48 h). DMSO used as a treatment control. Blots are representative of *n* = 2. (**b**) Densitometry analysis of Western blots. Relative protein expression levels were normalized to alpha tubulin expression.

## 4. Discussion

Rhabdoid tumors are rare but highly aggressive cancer types of early childhood [1]. Due to the lack of an effective conventional therapy regimen and the high toxicity of chemotherapy, patients suffering from rhabdoid tumors have poor overall survival rates (less than 50%) [11,12]. In order to improve treatment options and increase the chances of survival, novel targeted therapies must be investigated.

To the best of our knowledge, the expression and the role of the NK1R/SP complex in rhabdoid tumors has not been described so far. In this study, we present for the first time the expression of *TACR1* and *TAC1* in rhabdoid tumors. It has been shown that the *TACR1* expression pattern for hepatoblastoma does not correlate with clinical characteristics [39]. Similar to our recent study in hepatoblastoma, we here demonstrated for rhabdoid tumors that the expressions of *TACR1* and *TAC1* do not seem to correlate to parameters such as stage of disease, gender, and age of diagnosis. Hence, this conserved expression appears to be independent of any biological or clinical characteristics and highlights the broad applicability of NK1R-targeted therapies for treating pediatric rhabdoid tumors.

The NK1R antagonist aprepitant has been in clinical use against chemotherapy induced nausea and vomiting [40]. Furthermore, the antitumor effects of aprepitant have been described in several cancer types, including osteosarcoma, hepatoblastoma, and lung cancer [39,41,42]. It should be highlighted that the safety and the tolerability of aprepitant has been demonstrated in clinical trials, with high dose treatment leading to mild side

effects at most [24]. In this study, we investigated the cancer cell killing effects of aprepitant, and demonstrated that rhabdoid tumor growth is inhibited by blocking the NK1R, in vitro.

We were able to demonstrate the anti-tumor activity of aprepitant in three different rhabdoid tumor cell lines using at least two established procedures (e.g., MTT, FACS). To model the heterogeneity of rhabdoid tumors, which arise at different sites in the body, we used 2 AT/RT cell lines and 1 RTK cell line. All cell lines carry the characteristic loss-of-function INI1 mutation, thus closely reflecting the developing pediatric malignancies. Furthermore, in combination with a platin-derived cytostatic cisplatin, an additive effect was observed.

It has been shown in breast cancer and hepatoblastoma that the anti-tumor effect of aprepitant is induced by activation of apoptotic signaling pathways [19,43]. In order to unravel the mechanism of cell death in rhabdoid tumors, we investigated apoptosis by using both Annexin V staining and Western blot analysis. Similar to the previous studies, our data reveal increasing portions of late and early apoptotic cell populations upon aprepitant exposure. Using Western blot analysis, we could confirm the induction of apoptotic signal on protein level. More precisely, we observed cleaved products of PARP-1 in RT cells upon incubation with aprepitant. Altogether, these findings reveal that aprepitant exposure induces cell death by upregulating apoptotic signaling pathways.

New treatment regimens of RTs, such as intense multimodal therapy, have shown significant survival improvements [12]. However, due to considerable concerns over severe cytotoxicity and late adverse effects, further intensification of cytostatic compounds is not feasible. Therefore, novel therapeutic approaches are desperately needed. As such, just recently, Theruvath et al. presented an approach utilizing B7-H3-targeted chimeric antigen receptor (CAR) T cells for the treatment of AT/RTs [44], highlighting recent efforts to generate targeted therapies for the treatment of rhabdoid tumors. Our data beg the question of whether this novel target can potentially be included into more sophisticated future therapies, such as CAR T cell design. We have recently started to bring such efforts on the way, even though they are sophisticated, expensive and time consuming. Either way, we believe that it is worthwhile to further investigate the role of targeting the NK1R in the treatment of rhabdoid tumors.

Our approach presented here, on the other hand, is much more straightforward and merely involves the application of a small molecule already approved by the FDA, although for other indications. Also, it is important to note that in order to achieve antitumor effects to be triggered via NK1R-antagonists, much higher doses are expected to be needed compared to what is needed for the treatment of nausea and vomiting. It is generally understood that the doses of aprepitant for an anticancer effect must likely be 10-fold higher, which then would correspond to the µM doses used in the experiments presented here.

There exists some evidence that aprepitant will likely be tolerated in higher doses. For example, there have been several large prospective randomized clinical trials investigating the role of NK1R antagonists as antidepressants, with mixed results [45,46]. Importantly, the safety and tolerability of the NK-1 receptor antagonist aprepitant was demonstrated in a placebo-controlled trial in patients with moderate-to-severe major depression. At a dose of 300 mg/day (several times higher than for antiemesis), aprepitant was well tolerated, and no statistically significant difference in the frequency of adverse events was observed as compared to placebo [45].

Further, we are currently using aprepitant as an off-label drug for some (desperate) cases in children with hepatocellular carcinoma or hepatoblastoma, although to this point not for rhabdoid tumors. We use doses five times higher than what is normally given to children as an antiemetic, in general up to 10 mg/kg or higher. In our case, we did not see general side effects that we could attribute to the NK1R-antagonist alone, with one important exception. Children on whom we had performed a liver transplant (for hepatoblastoma) and who are on immunosuppression show a strong interaction with their tacrolimus levels, which in some cases immediately become sky high and can trigger kidney disfunction. This reaction is likely due to inhibition of the cytochrome p450 system.

Therefore, caution is warranted in children who are on tacrolimus and related medications when giving aprepitant in high doses.

There are several limitations to our study. The patient cohort (*n* = 43) of the analyzed database is rather small. Given that the age-standardized incidence for extracranial rhabdoid tumors and for AT/RTs in children is 0.6 per 1 million and 0.07 per 100,000, respectively, this small sample size reflects the rarity of the disease. Nevertheless, we confirmed our findings using primary patient material obtained from children treated at our hospital. Again, we could only obtain a few tissue samples, but we argue that the number is sufficient to allow for further investigation of our hypothesis [47,48]. Also, we did not correlate our findings in two liver tumor specimens with corresponding cell lines. The reason is that, different from the two other organ systems presented, no viable cell line exists for rhabdoid tumors of the liver. Furthermore, in our study, we do not investigate the in vivo efficacy of the treatment. However, the efficacy of aprepitant treatment in cancer cells has already been presented in experimental mouse models for other tumors [19,42].

## 5. Conclusions

For the first time, we present the expression of *TACR1* and *TAC1* in rhabdoid tumors, the potent inhibitory effect of the NK1R antagonist aprepitant in this tumor entity and the analysis of apoptotic pathways of rhabdoid tumor cells upon treatment with aprepitant. Our results strongly suggest further studies of the NK1R/SP complex and its antagonist in rhabdoid tumors to implement this novel therapeutic approach in its treatment.

**Supplementary Materials:** The following are available online at https://www.mdpi.com/article/10.3390/curroncol29010008/s1. Figure S1: Determination of apoptotic cell populations of G-401, BT-12, CHLA-266 and HepG2 (second representative); Figure S2: Determination of apoptotic cell populations of G-401, BT-12, CHLA-266 and HepG2 (third representative); Figure S3: Western blot analysis of G-401, BT-12 and CHLA-266 upon treatment with aprepitant and densitometry analysis of Western blots (first representative); Figure S4: Western blot analysis of G-401, BT-12 and CHLA-266 upon treatment with aprepitant and densitometry analysis of western blots (second representative).

**Author Contributions:** Conceptualization, M.B.; methodology, M.B., S.D., A.G., J.K. and R.K.; validation, J.K., S.D. and A.G.; formal analysis, J.K., S.D., A.G.; investigation, J.K., S.D., A.G., D.L.; resources, R.K., M.M.D., C.W., D.v.S., T.M., F.H., S.K.; data curation, J.K., S.D., A.G.; writing—original draft preparation, J.K. and S.D.; writing—review and editing, M.B., S.D., R.K., S.K., A.G., M.I. and I.B.; visualization, J.K., S.D., A.G. and D.L.; supervision, M.B.; project administration, M.B.; funding acquisition, M.B., R.K. and S.K. All authors have read and agreed to the published version of the manuscript.

**Funding:** This research was funded by the "Promotionsstudium Förderung für Forschung und Lehre" program from the University of Munich, LMU (to J.K. and M.B.). M.B. was also funded by the Friedrich-Baur Institute. Additionally, this study was supported by the Marie-Sklodowska-Curie Program Training Network for the Immunotherapy of Cancer funded by the H2020 Program of the European Union (Grant 641549, to S.K.), the Marie-Sklodowska-Curie Program Training Network for Optimizing Adoptive T Cell Therapy of Cancer funded by the H2020 Program of the European Union (Grant 955575, to S.K.), the Hector foundation, the International Doctoral Program i-Target: Immunotargeting of Cancer funded by the Elite Network of Bavaria (S.K.); Melanoma Research Alliance Grants 409510 (to S.K.); the Else Kröner-Fresenius-Stiftung (S.K.); the German Cancer Aid (S.K.); the Ernst-Jung-Stiftung (S.K.); LMU Munich's Institutional Strategy LMUexcellent within the framework of the German Excellence Initiative (to S.K.); the Bundesministerium für Bildung und Forschung Project Oncoattract (to S.K.); by the European Research Council Grant 756017, ARMOR-T (to S.K.), by the German Research Foundation (DFG to S.K.), the Fritz-Bender Foundation (to S.K.) and the José-Carreras Foundation (to S.K.). Fabian Hauck is funded by the Care-for-Rare Foundation (C4R, 160073), the Else Kröner-Fresenius Stiftung (EKFS, 2017_A110), and the German Federal Ministry of Education and Research (BMBF, 01GM1910C).

**Institutional Review Board Statement:** The study was conducted according to the guidelines of the Declaration of Helsinki and approved by the Institutional Review Board (or Ethics Committee) of Ludwig-Maximilians-University Munich, Germany (protocol code 19-115 and 21-0191).

**Informed Consent Statement:** Informed consent for surgical tissue sampling was obtained from all subjects involved in the study. The tissue samples used in this study represent leftover tissue from previous diagnostic workup. Given that tissue samples date back as far as 20 years from the original sampling and that tissue samples were available in an anonymous format, additional informed consent was waived.

**Data Availability Statement:** The data presented in this study are openly available in the cBio Cancer Genomics Portal (http://cbioportal.org, accessed on 2 December 2021) [28,29]. Five pediatric cancer studies on acute myeloid leukemia (AML; phs000465), acute lymphoblastic leukemia (ALL; phs000464), neuroblastoma (NB; phs000467), rhabdoid tumors (RT; phs0004709), and Wilms tumor (WT; phs000471) were selected from the Therapeutically Applicable Research to Generate Effective Treatments (https://ocg.cancer.gov/programs/target, accessed on 2 December 2021) initiative and mRNA expression of 2 genes was analyzed (*TACR1*, *TAC1*).

**Acknowledgments:** The authors would like to thank A. Hotes and T. Schmidt for excellent technical assistance. We thank the iFlow Core Facility of the university hospital Munich for assistance with the generation of flow cytometry data. We thank T. Magg and F. Hauck for providing the fibroblast cell lines.

**Conflicts of Interest:** S.K. has received honoraria from Novartis, TCR2 and GSK. S.K. is an inventor of several patents in the field of immuno-oncology. S.K. received research support from TCR2 Inc and Arcus Bioscience for work unrelated to this manuscript.

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
