# Peer review of "The Neurokinin-1 Receptor Is a Target in Pediatric Rhabdoid Tumors"

_curroncol, doi:10.3390/curroncol29010008_

Round 1

Reviewer 1 Report

The manuscript of original research by Kolorz, et al. titled “The Neurokinin-1 receptor is a target in pediatric rhabdoid tumors” describes the possible therapeutic role of a neurokinin-1 receptor antagonist, aprepitant, for rhabdoid tumors (RT) which is one of the rarest malignancies in childhood whereas very aggressive behavior. Although development of novel and effective treatment for RT has been pursued by numbers of pediatric oncologists, it was extremely hard because of the rarity of the disease. Therefore, this manuscript should be tremendously important in pediatric oncology. The design and methods of the experiments are satisfactory with their logical approach to disclose the role of neurokinin-1 receptor/ substance P complex in RT. Moreover, the results showed objective efficacy of aprepitant, which is the FDA-approved drug and actually used in combination with other anti-cancer agents such as cisplatin. This story is quite fascinating for pediatric oncologists like me. I believe that these results are worth being published in Current Oncology.

Please consider to make minor amendments to improve the manuscript further.

  1. In abstract (line 34), an abbreviation “NK1R” firstly appears. Please describe its full spelling.
  2. Table 2 contains “*” but no description to explain this. Please add the explanation.
  3. Please improve a configuration of the manuscript including a layout of the figures and the tables, although the editorial office might be able to work on this matter.

Author Response

The manuscript of original research by Kolorz, et al. titled “The Neurokinin-1 receptor is a
target in pediatric rhabdoid tumors” describes the possible therapeutic role of a neurokinin-1
receptor antagonist, aprepitant, for rhabdoid tumors (RT) which is one of the rarest
malignancies in childhood whereas very aggressive behavior. Although development of novel
and effective treatment for RT has been pursued by numbers of pediatric oncologists, it was
extremely hard because of the rarity of the disease. Therefore, this manuscript should be
tremendously important in pediatric oncology. The design and methods of the experiments
are satisfactory with their logical approach to disclose the role of neurokinin-1 receptor/
substance P complex in RT. Moreover, the results showed objective efficacy of aprepitant,
which is the FDA-approved drug and actually used in combination with other anti-cancer
agents such as cisplatin. This story is quite fascinating for pediatric oncologists like me. I
believe that these results are worth being published in Current Oncology.
Please consider making minor amendments to improve the manuscript further.
1. In abstract (line 34), an abbreviation “NK1R” firstly appears. Please describe its full
spelling.
2. Table 2 contains “*” but no description to explain this. Please add the explanation.
3. Please improve a configuration of the manuscript including a layout of the figures and
the tables, although the editorial office might be able to work on this matter.

We thank the reviewer very much for his or her kind remarks. We changed the abbreviation
for NK1R in line 34, added the explanation for “*” in Table 2. Concerning the layout
configuration for the figures and the tables we improved it and will work with the editorial office
in the future to perfect it.

Reviewer 2 Report

This is a neat study testing the levels of neurokinin-1 receptor in pediatric rhabdoid cancer and preclinical data showing the benefit of combining neurikinin-1 inhibitor with cisplatin to reduce cancer cell viability.

The only minor concern I have is regarding the high concentration of aprepitant needed for suppressing rhabdoid cell viability. Could the authors comment on whether it is possible to achieve such high concentrations in the patient?

Author Response

This is a neat study testing the levels of neurokinin-1 receptor in pediatric rhabdoid cancer
and preclinical data showing the benefit of combining neurikinin-1 inhibitor with cisplatin to
reduce cancer cell viability.
The only minor concern I have is regarding the high concentration of aprepitant needed for
suppressing rhabdoid cell viability. Could the authors comment on whether it is possible to
achieve such high concentrations in the patient?

We thank this reviewer for his or her comments. The question raised is intriguing and we have
asked it ourselves many times. This point is especially important since it is known that doses
of aprepitant would have to be significantly higher to create an antitumor effect compared to
the doses given for, for example, antiemesis. NK1R antagonists are approved for clinical use
by the Food and Drug Administration (FDA) in the form of the compound aprepitant (Emend®)
as an antiemetic medication since 2003. Aprepitant is a small molecule with a molecular mass
of 534,43 g/mol. From a clinical perspective, aprepitant was first described to have a particularly strong effect in nausea and vomiting that is associated to chemotherapy. There
have been several large prospective randomized clinical trials investigating the role of NK1R
antagonists as antidepressants, with mixed results. Importantly, safety and tolerability of the
NK-1 receptor antagonist aprepitant was demonstrated in a placebo- controlled trial in patients
with moderate-to severe major depression. At a dose of 300mg/day (several times higher than
for antiemesis), aprepitant was well tolerated, and no statistically significant difference in the
frequency of adverse events was observed as compared to placebo. Also, there is strong
evidence accumulating recently from many prospective randomized controlled clinical trials
showing efficiency of NK1R inhibitors in postoperative nausea and vomiting in both adults and
children for a large variety of different surgeries. No NK1R antagonist to date is approved
specifically for cancer therapy, and there are currently no ongoing clinical trials that assess
their therapeutic potential regarding cancer.

Indeed, currently and ongoing we are using aprepitant as an off-label drug for some
(desperate) cases in which the patients were children with hepatocellular carcinoma or
hepatoblastoma, although to this point not for rhabdoid tumors. We used doses 5 times higher
than what is normally given to children as an antiemetic, in general up to 10mg/kg or higher.
In our hands we did not see general side effects that we could attribute to the NK1R-antagonist
alone, however, these are complex patients with advanced disease receiving a multitude of
medications, including aggressive chemotherapy. It is therefore near impossible to identify
each and every attributable (side)effect. Having said that, there is one particularly important
exception. We treated one child on who we had performed a liver transplant and who was on
immunosuppression. This child had recurrent pulmonary metastasis unresponsive to chemo
and surgery, and we treated him with high doses of aprepitant. We saw a strong interaction
with the tacrolimus levels, which immediately became sky high and triggering kidney
disfunction. This reaction was likely due to inhibition of the cytochrome p450 system, which is
described. Therefore, caution is warranted in children who are on tacrolimus and related
medications when giving aprepitant in high doses.

When giving aprepitant to children, it is important to consider its pharmacokinetic properties.
In the past, we have been in contact with MSD Sharpe and Dohme GmbH, which makes
aprepritant and markets it as the antiemetic drug Emend®. According to them, as well as per
their official data sheet (available at
https://www.accessdata.fda.gov/drugsatfda_docs/label/2005/021549s009lbl.pdf), they state
on page 2 under the section “pediatric” that “The pharmacokinetics of EMEND have not been
evaluated in patients below 18 years of age.” Nevertheless, there are now accumulating some
studies in the literature assessing the pharmacokinetic properties of aprepitant in children
given for antiemesis. A recent paper by Salman et al. provides level I evidence as part of a
randomized, controlled trial and is a great example in this regard 3. In their paper, of 220
randomized and treated children undergoing chemotherapy, 119 receiving a single aprepitant
dose as an antiemetic were sampled for PK analysis and had evaluable aprepitant plasma
concentrations. A dose-dependent relationship in exposure (AUC0-8 h and Cmax) was
observed. Aprepitant was generally well tolerated, and the CR and NV rates were high (>80%)
across treatment groups.

Another important study in this regard is that of Mora et al. published recently in Pediatric
Blood and Cancer 4. Their analysis is similar in setup compared to that of Salman et al. described above with the exception that fosaprepitant was given for antiemesis instead of
aprepitant. Fosaprepitant, different from aprepitant, is given intravenously, which is appealing
in a clinical setting because in a potential cancer therapy, some children many have a
compromised GI tract and might not tolerate oral medications. In their study, PK data were
evaluable for 167/234 children. Aprepitant exposures were dose proportional; adolescents (12
to 17 years) receiving fosaprepitant 150 mg had exposures similar to adults at the same dose.
The adverse event profile was typical of cancer patients receiving emetogenic chemotherapy.
Drug-related adverse events were reported in 16 (6.8%) subjects, with hiccups being most
common (n = 5; 2.1%). These data support that fosaprepitant could be a suitable compound
to use for potential anticancer therapies in case an oral therapy is not tolerated or is otherwise
not suitable.

A third study important in this regard is that of Chain et al. recently published in The Journal
of Pharmacology and Therapeutics 5. Their analysis included 1326 aprepitant plasma
concentrations from 147 patients. Aprepitant PK was described by a 2-compartment model
with linear elimination and first-order absorption, with allometric scaling for central and
peripheral clearance and volume using body weight, and a cytochrome P450 3A4 maturation
component for the effect of ontogeny on systemic clearance. Their simulations recognized
that the application of a weight-based (for those <12 years) and fixed-dose (for those 12–17
years) dosing regimen results in comparable exposures to those observed in adults. This last
part is interesting as it is in accordance with what was found by Mora et al., which was that
higher weight-normalized doses (5 mg/kg) were necessary for children aged < 12 years to
achieve comparable adult exposures.

Therefore, all these recent reports on the pharmacokinetics of aprepitant in children hint that
aprepitant would have to be given orally and at least daily in order to reach sufficient serum
levels and this could be carried out sufficiently and efficiently. Similarly, fosaprepitant would
have to be given also daily and could be applied intravenously. Children under the age of 12
years would likely need higher weight-normalized doses compared to adults, and children
above age 12 may be treated with a fixed dose.

It must be kept in mind, though, and this is our biggest concern at this time, that all of these
studies were carried out in children undergoing treatment with aprepitant and fosaprepitant for
antiemesis. It is generally understood that the doses of aprepitant for an anticancer effect must
likely be 10-15 fold higher (which then would correspond to the μM doses used in the
experiments presented) compared to what is given as an antiemetic. Therefore, at this time it
is unclear whether the emerging knowledge of the pharmacokinetic properties of aprepitant
when used in the low doses for antiemesis are expandable to the child’s physiology when
given in the significantly higher doses required for anticancer therapy.

For these and other reasons, and like many other researchers in the field of NK1R targeting,
we caution to idealize aprepitant, as it is available today as an FDA approved drug, to be the
sole and perfect compound used in future anticancer strategies. Rather, either different
compounds with higher affinity to or longer duration at the NK1R would likely be more suitable,
or, as we are exploring experimentally as of this time in our laboratory for hepatoblastoma
cells, a CART cell targeting tumor cells via the NK1R are more fitting (or perhaps a
combination of treatments).

Reviewer 3 Report

Kolorz et al report a study of expression patterns and preclinical therapeutic targeting of the neurokinin-1 receptor in rhabdoid tumor.  This is a novel target in an aggressive, rare malignancy and potentially opens new translational opportunities.  The study is scientifically very well done and clearly described.  I have one major and then several minor comments and do recommend publication.

Major comment:

Fig 2, 3, and 4.  Why are concentration ranges given for aprepitant after Fig 2A?  None of these experiments are titrations so there should be only one dose.  This needs to be clearly described and, if not a typo, the variation should be justified.

Minor comments:

Page 2, lines 62-66.  These sentences are very long and could benefit from editing to be more concise.

Page 2, line 72.  Is there a typo perhaps?  Saying upregulation of the truncated isoform is inversely related to rapid growth sounds backwards from the rest of the descriptions.

Fig 1b. The lack of statistical significance is a little surprising given the visible differences between eg RT and WT.  It is true the error bars are large so perhaps this is true, but I suggest double checking the statistical tests.

Page 5, line 202-203.  It says HepG2 is known to have high expression of TACR1-fl but in Fig 1c it seems to be background.  Why are these inconsistent?

Table 2. The asterix needs explicit explanation.

Table 3.  This table is confusing.  This seems to be a description of the cBioportal/TARGET patient cohort so I don't understand why there are different columns for TACR1 and TAC1. 

Page 9, lines 285-291.  The authors neglect to describe Fig2A (aprepitant monotherapy) results.  This should be included before describing Fig 2B.

Author Response

Kolorz et al report a study of expression patterns and preclinical therapeutic targeting of the
neurokinin-1 receptor in rhabdoid tumor. This is a novel target in an aggressive, rare
malignancy and potentially opens new translational opportunities. The study is scientifically
very well done and clearly described. I have one major and then several minor comments
and do recommend publication.
Major comment:
Fig 2, 3, and 4. Why are concentration ranges given for aprepitant after Fig 2A? None of
these experiments are titrations so there should be only one dose. This needs to be clearly
described and, if not a typo, the variation should be justified.

We thank the reviewer for his or her remarks. Concerning the major comment that the reviewer
had, we agree and added the dose that was used in each experiment.

Minor comments:
Page 2, lines 62-66. These sentences are very long and could benefit from editing to be
more concise.
Page 2, line 72. Is there a typo perhaps? Saying upregulation of the truncated isoform is
inversely related to rapid growth sounds backwards from the rest of the descriptions.
Fig 1b. The lack of statistical significance is a little surprising given the visible differences
between eg RT and WT. It is true the error bars are large so perhaps this is true, but I
suggest double checking the statistical tests.
Page 5, line 202-203. It says HepG2 is known to have high expression of TACR1-fl but in
Fig 1c it seems to be background. Why are these inconsistent?
Table 2. The asterix needs explicit explanation.
Table 3. This table is confusing. This seems to be a description of the cBioportal/TARGET
patient cohort so I don't understand why there are different columns for TACR1 and TAC1.
Page 9, lines 285-291. The authors neglect to describe Fig2A (aprepitant monotherapy) results. This should be included before describing Fig 2B.

Concerning the minor comments of the reviewer we rearranged the long sentences in lines
62-66 and changed the typo in line 72. We double checked the statistical tests, however,
unfortunately the results stayed the same that the error bars are so large, which explains the
lack of statistical significance. On Page 5 we changed the sentence; added an explicit
explanation for the asterix in table. Concerning your remarks about table 3, there was one
patient in the cohort that had only staging and TAC1 expression information. This explains the
difference between the two columns. The description of figure 2a is now marked and improved
in the paper.

This manuscript is a resubmission of an earlier submission. The following is a list of the peer review reports and author responses from that submission.

Round 1

Reviewer 1 Report

Targeting neurokinin-1 receptor by aprepitant was described by the same authors in several pediatric tumors in the last decade. Now an another tumor type was described. The drug is in our hand for human use. Next important step with aprepitant would be to show the results of a human study.

As Aprepitant is already in use as an antiemetic, all the pharmacokinetic data are in hand. The effective concentrations of this study should be compared to PK data of aprepitant, to find out, if there is any rationality to use this drug as anticancer drug.

Reviewer 2 Report

The study is of interest, the methods and results are well described and discussed. I do not have questions or suggestions to contribute. 

Author Response

Thank you very much for your kind remarks.

Reviewer 3 Report

congratulations for this original work, openique new perspespectives for the therapeutic approach of Rabdoid Tumors

Author Response

(The authors gave the same response as above.)

Reviewer 4 Report

In this manuscript, the authors report the expression of a splicing variant truncated neurokinin-1 receptor (NKR1; TACR1) in rhabdoid tumor-derived cells, as a proposed cancer associated mechanism, and explore targeting this pathway. Overall, there are major flaws in the results and experimental designs, and the major defect is that the study shows correlational results and no evidence of causation is shown, and that many of the results presented are not convincing. • Although the results showing a prominent expression of the truncated TACR1 variant in the tested RT cells over the full-length version seem robust, there is lack of negative controls in this experiment. • The analysis from human databases is indirect, as only effector TAC1 expression levels are shown, and the expression of truncated TACR1. The comparison of RB with AML, ALL and NB and not other CNS tumors (such as GBM/pHGG) does not seem to be clearly justified. I suggest the authors to explore tools to analyze alternative splicing from this or other databases’ RNA-seq data. • The experiment to analyze the sensitivity to the NK receptor agonist are not clear. Even at the highest dose of inhibitor aprepitant, the IC50 is not reached for any of the RB cells tested. The positive effect of the combination with cisplatin is not clear either, as G-401 and BT-12 cells do not show a significant difference between Apre and Cis + Apre treatments. The effect of SP to reverse aprepitant inhibition is not convincing, only being significant in one out of the three RB cells tested. Finally, these experiments lack negative controls with cells that do no express truncated TACR1. • There is a lack of a statistical analysis for the flow cytometry-based apoptosis analysis. Frequencies of apoptotic/live cells of the three experimental repeats should be combined • Genetic engineering of the RB cell models to inhibit/overexpress truncated/full TACR1 should be included to demonstrate the association between truncated TACR1 and susceptibility to NK receptor agonist.

Reviewer 5 Report

In the manuscript by Kolorz et al. titled, "The Neurokinin-1 receptor is a target in pediatric rhabdoid tumors," the authors present the novel application of a NK1R inhibitor to treat RT. This is the first application of this targeted therapy in RT and while it generates some excitement as to the immediate use of an FDA-approved NK1R inhibitor, the effects of inhibition are modest. The authors accurately state their limitations as well as pointing the reader to the necessity of combination therapies for efficacy. The authors do overstate their conclusion; however, on page 10 line 301 that they used conventional cytostatic drugs, when in reality they only used one, cisplatin. Also, I would recommend stating that "NK1R may be a promising target for the treatment of RT in combination with other anti-cancer therapies..."

The presentation of the overall survival differences between high and low expression of TACR1 and TAC1 further underlines the insignificance of going after this target. Additionally, I have issues with the cell viability studies in Figure 2A. The authors measured IC50s for the different cell lines; however, for three of the cell lines the curves never cross below the 50% viability threshold. 

Lastly, the authors state in their methods section multiple times that they were seeding 96 well plates with 500,000 cells/well (5x105). This cannot be right. It is 2 orders of magnitude more than the typically cell viability experiment for a 96 well plate. The authors also never reference their supplemental data in the text. 

Round 2

Reviewer 1 Report

Since my last read, the paper did not change significantly. This paper in novelty, significance of content and completeness does not merit the level of Cancers

Reviewer 4 Report

The authors provided convincing responses on the concerns pointed, although this is not reflected on deep changes on the manuscript, e.g., the IC50 for aprepitant was not reached and the IC50 was extrapolated, and the results regarding this experiments are questionable. The responses are accurate, e.g., we used concentrations of aprepitant that were lower than the IC50, I do not have any comment to offer, I will leave the decision to the consideration of the editors. The only main concern about the revision is that the statistical analysis on the flow cytometry experiment is lacking, i.e., P values among the different compared experimental groups. 

Reviewer 5 Report

The authors have addressed the previous concerns. At present, it is acceptable for a broader viewership.